# The Sleeping Brain: Harnessing the Power of the Glymphatic System through Lifestyle Choices

**DOI:** 10.3390/brainsci10110868

**Published:** 2020-11-17

**Authors:** Oliver Cameron Reddy, Ysbrand D. van der Werf

**Affiliations:** Department of Anatomy and Neurosciences, Amsterdam UMC, Vrije Universiteit Amsterdam, Amsterdam Neuroscience, De Boelelaan, 1117 Amsterdam, The Netherlands; reddy.oliver@gmail.com

**Keywords:** glymphatic system, protein aggregates, Alzheimer’s disease, amyloid-beta, sleep, disease prevention

## Abstract

The glymphatic system is a “pseudo-lymphatic” perivascular network distributed throughout the brain, responsible for replenishing as well as cleansing the brain. Glymphatic clearance is the macroscopic process of convective fluid transport in which harmful interstitial metabolic waste products are removed from the brain intima. This paper addresses the glymphatic system, its dysfunction and the major consequences of impaired clearance in order to link neurodegeneration and glymphatic activity with lifestyle choices. Glymphatic clearance can be manipulated by sleep deprivation, cisterna magna puncture, acetazolamide or genetic deletion of AQP4 channels, but how lifestyle choices affect this brain-wide clearance system remains to be resolved. This paper will synthesize existing literature on glymphatic clearance, sleep, Alzheimer’s disease and lifestyle choices, in order to harness the power of this mass transport system, promote healthy brain ageing and possibly prevent neurodegenerative processes. This paper concludes that 1. glymphatic clearance plays a major role in Alzheimer’s pathology; 2. the vast majority of waste clearance occurs during sleep; 3. dementias are associated with sleep disruption, alongside an age-related decline in AQP4 polarization; and 4. lifestyle choices such as sleep position, alcohol intake, exercise, omega-3 consumption, intermittent fasting and chronic stress all modulate glymphatic clearance. Lifestyle choices could therefore alter Alzheimer’s disease risk through improved glymphatic clearance, and could be used as a preventative lifestyle intervention for both healthy brain ageing and Alzheimer’s disease.

## 1. Introduction

Discovered in 2012, the glymphatic system, which stands for glial-dependent lymphatic transport, has been categorized as a macroscopic waste clearance system. Due to the similarities in function, the glymphatic system has been described as the central nervous system’s analogue to the lymphatic system [1,2]. The transportation of the central nervous system’s interstitial fluid (ISF) has long been thought to move via diffusion, but recently ISF was observed moving at a much faster rate than that possible through diffusion. This suggests the involvement of a mass transport system [3]. This glial cell-dependent paravascular network removes soluble proteins and metabolites from the central nervous system, but in addition supplies the brain with glucose, lipids and neuromodulators, utilizing paravascular tunnels formed by astroglial cells [1,2,4]. Since this is a relatively new discovery, the amount of scientific literature surrounding the glymphatic system is rapidly increasing, and therefore its definition is continuously being renewed. This has caused controversy surrounding both the directionality and the anatomical space in which this system resides. For instance, the movement of ISF along paravascular spaces of veins remains disputed, and some claim that a distinct route exists for this clearance pathway [5]. These discrepancies can, however, be partially explained by the limited amount of literature and methodological differences between studies [1,5].

The glymphatic system is constantly filtering toxins from the brain, but during wakefulness, this system is mainly disengaged [1]. During natural sleep, levels of norepinephrine decline, leading to an expansion of the brain’s extracellular space, which results in decreased resistance to fluid flow. This is reflected by improved cerebrospinal fluid (CSF) infiltration along the perivascular spaces, and therefore increased interstitial solute clearance [2]. The increase in clearance happens specifically during non-rapid eye movement sleep (N), also known as quiescent sleep. The third N stage, N3 or slow-wave sleep, is categorized by slow oscillatory brain waves, that create a flux of CSF within the interstitial cavities, leading to an increase in glymphatic clearance [6,7,8]. The role of sleep in glymphatic clearance has been conclusively demonstrated, and since the vast majority of clearance occurs during sleep, the glymphatic system can simply not be investigated without examining the basic aspects of sleep.

Impaired glymphatic clearance has been linked to neurodegenerative diseases [1]. Alzheimer’s disease is a chronic neurodegenerative disease and the most common dementia, typically beginning with disorientation and then proceeding to a gradual deterioration of memory, language and physical independence, among others [1]. Amyloid-beta and tau protein aggregations are heavily associated with Alzheimer’s disease, creating plaques and neurofibrillary tangles in the brain that lead to brain degradation [2,3]. Glymphatic clearance moves tau proteins and amyloid-beta aggregates out of the brain [1,3]. This suggests that the glymphatic system is involved in modulating, or possibly protective against, Alzheimer’s disease. This paper will focus on Alzheimer’s disease, since it is the most frequent dementia, but will hopefully remain applicable to other neurodegenerative diseases, since several dementias are thought to be caused by protein aggregation. The need for an intervention is gaining urgency [1,2,3,4]. Benveniste and colleagues recently used MRI scans in combination with contrast agents to monitor CSF flow through the brain in real time [1], yet a method for manipulating glymphatic activity in humans still remains to be developed. Regulating glymphatic clearance could increase waste removal of aggregates in diseases associated with protein deposition, slowing or even reversing neurodegeneration.

Sleep is a primary driver of glymphatic clearance. However, research on a wealth of other lifestyle choices such as sleep quality, quantity, physical exercise, changes in body posture, omega 3, chronic stress, intermittent fasting and low doses of alcohol has begun to emerge. Despite these advances, scholars in this field have not yet adequately harnessed the power of lifestyle-regulated glymphatic clearance. Lifestyle choices remain to be evaluated and compared. No guides or literature reviews exist on how to use preventative measures to bolster glymphatic activity. With the incidence of neurodegenerative disease increasing and evidence of the glymphatic systems’ involvement growing, there is an urgent need to capitalize on the uses of this mass transport system. Lifestyle changes decelerating disease progression could be an important discovery, opening a therapeutic avenue and the potential for improvements in quality of life.

In order to infer the causal relationships of lifestyle choices in reducing brain ageing and Alzheimer’s disease, this paper will first investigate why glymphatic clearance primarily occurs during sleep, and which underlying mechanisms drive glymphatic clearance. Next, this paper will inspect the implications of a dysfunctional glymphatic pathway and establish the relationship between glymphatic clearance and neurodegenerative disease. Finally, this paper will investigate how lifestyle choices affect this mass transport system and how they can be used as a protective and preventive measure in the context of aging and Alzheimer’s disease.

## 2. Materials and Methods

This literature review synthesizes research on the glymphatic system, neurodegenerative disease and various lifestyle choices. By critically analyzing this literature, we aim to work towards a preventive guide for Alzheimer’s disease, and hopefully also other neurodegenerative diseases. We have gathered papers from the scientific database “PubMed”, as well as additional sources from the reference lists of some of the papers found through the database searches. Using the keywords “glymphatic” and “system”, the search yielded 389 results. Of these 389 papers, those including lifestyle choices and those related to Alzheimer’s disease were selected, based on title, abstract and applicability to the research question. A PRISMA flow chart documents the precise selection process and assessment of eligibility (see Figure 1).

## 3. Results

### 3.1. The Glymphatic System of the Brain

#### 3.1.1. Fluid Movement in the Brain

The brain consists of four aqueous compartments: CSF, ISF, intracellular fluid and blood, all separated by two main barriers regulating their ionic and biochemical composition: the blood–brain barrier and the blood–CSF barrier [2] (see Figure 2). The blood–brain barrier is located throughout the brain along the vasculature. Tight junctions on endothelial cells block the movement of macromolecules but allow fluids and solutes to diffuse into the brain from the perivascular space between endothelial cells and astrocytic endfeet [2]. The blood–CSF barrier, on the other hand, has fenestrated endothelial cells allowing macromolecules into the interstitial space. This barrier is located within the choroid plexus of the two lateral, the third and the fourth ventricles. Its epithelial cells have an abundance of tight junctions in order to regulate CSF composition [1]. These do allow the movement of macromolecules and principally transport Na, K, Cl and HCO_3_ ions through primary active transport using a Na/K ATPase [2]. The constant production of CSF by the choroid plexus drives the flow direction of CSF through the brain [1]. Collectively, this results in a CSF production of around 500 mL each day, flowing from the lateral ventricles to the third and then fourth ventricle, entering the subarachnoid space, bathing the brain. It exits predominantly through the perineural spaces of the cranial nerves along the internal carotid artery, or into the olfactory-nasal submucosa pathway, ultimately draining into deep cervical lymph nodes [1,2].

#### 3.1.2. Paravascular Spaces

Within the interstitial spaces of the brain, CSF travels towards perivascular and perineural spaces, and in doing so clears solutes from the neuropil into meninges. These then exit the brain and drain into cervical lymphatic vessels and are ultimately degraded in the liver [3]. CSF enters the brain via para-arterial channels and exchanges with ISF, which in turn is cleared by paravenous pathways (depicted in Figure 3.) [4]. CSF from the subarachnoid spaces enters the Virchow–Robin spaces along para-arterial channels and exchanges with ISF, contrary to the classical model of CSF secretion and absorption [4]. CSF enters the brain exclusively via periarterial spaces and ISF drains exclusively into perivenous spaces [2]. The CSF influx is balanced by the perivenous efflux of ISF ridding the neuropil of proteinaceous metabolites. Patients suffering from central oedema show a significant decrease in CSF entering perivascular spaces, suggesting that a CSF movement is driven not only by a pressure difference, but also by pulsations of arterial smooth muscle [1,2].

#### 3.1.3. Fluid Movement within the Interstitial Space

Once deeper within the brain, CSF movement is facilitated by aquaporin 4 (AQP4) water channels on the endfeet of astrocytes, which ensheathe the blood vasculature. CSF then enters the parenchyma and mixes with the ISF, where both continuously interchange [3]. The separation here has been proven by AQP4 knockout mice, which have significantly less CSF to ISF exchange than wild type mice, suggesting that AQP4 channels are responsible for CSF–ISF exchange, but the influx of CSF into the periarterial spaces was not affected [2]. Although convection occurs within the perivascular spaces, within the extracellular space, diffusion is responsible for the movement of ISF. Overlapping astrocytic endfeet which completely ensheathe the cerebral microvasculature inhibit the access of molecules with larger molecular weight from entering the interstitium [4].

### 3.2. Sleeping the Brain Clean

“Innocent sleep. Sleep that soothes away all our worries. Sleep that puts each day to rest. Sleep that relieves the weary labourer and heals hurt minds. Sleep, the main course in life’s feast, and the most nourishing”―William Shakespeare, Macbeth (2.2.50–52).

In 1606, William Shakespeare was already aware that sleep has vital and specific roles: to repair both the body and mind. Four hundred years later, sleep still largely remains an enigma and is one of the last physiological processes with a lack of scientific consensus [10]. What we do know is that sleep is a quiescent behavioral state, associated with reduced responsiveness to weak stimuli and rapid reversibility in response to strong stimuli, and is required for memory formation, brain plasticity and immune function, among others [11]. Sleep comes in two metabolic and electrophysiological varieties, namely rapid eye movement sleep (R) and non-rapid eye movement sleep (N) [11]. In order to correctly classify these stages, this paper will use the American Academy of Sleep Medicine (AASM) scoring, classifying sleep stages as N1, N2, N3 and R sleep, replacing the previous Rechtschaffen and Kales 1968 scoring of sleep stages: NREM 1, 2, 3, 4 and REM sleep [12]. Regardless of the classification, the N sleep stages are not considered distinct entities, rather gradual transitions in wave form densities detected by electroencephalography [12].

#### 3.2.1. The Glymphatic System and Sleep

The glymphatic system is constantly filtering toxins from the brain, but during wakefulness, this system remains mainly disengaged [3]. Although sleep is often associated with rest, glymphatic activity is dramatically boosted during sleep. Photoimaging of in vivo mice demonstrated a 90% reduction in glymphatic clearance during wakefulness, and twice the amount of protein clearance from the brain intima during sleep [1]. Sleep-induced enhancement of glymphatic function appears to arise from the expansion of the ISF space [13]. In a human in vivo study, blood oxygen level-dependent functional magnetic resonance imaging (Bold fMRI) was used in combination with electroencephalograph and CSF measurements in order to detect in which sleep state most brain activity occurred. They found that during wakefulness, CSF flow had a small-amplitude rhythm, peaking at around 0.25 HZ, whereas during sleep, large oscillations occurred every 20 s, peaking at around 0.05 HZ, resulting in a significantly greater inflow of CSF than during the day [6]. As well as cleansing the brain, the replenishing role of the glymphatic system was observed. Glymphatic-induced reoxygenation of the brain occurs during large pulsations of CSF. The pulsating fashion in which these sleep oscillations occur suggests that the majority of glymphatic activity occurs during N3 sleep. During this stage of sleep, slow oscillatory brain waves were shown to increase the amount of CSF within the interstitial cavities, leading to an 80–90% increase in glymphatic clearance relative to the waking state, and demonstrate the importance of slow-wave sleep [6,7,8].

#### 3.2.2. Slow-Wave Sleep

Glymphatic clearance mainly occurs in slow-wave sleep, which is synonymous with the N3 sleep stage. It is characterized by high-voltage synchronized electroencephalograph waveforms: delta oscillations and slow oscillations [13]. Slow-wave sleep has numerous functions including learning, memory and metabolite clearance [12]. In young adults, slow-wave sleep makes up between 10 and 25% of total sleep time, but this kind of sleep is not evenly distributed throughout the night, mostly occurring in the first half, with more R sleep occurring in the second half [12]. These slow waves usually lie between 0.5 and 4.5 hertz on electroencephalography and have been linked to sleep pressure, occurring in abundance early in the night and then decreasing [7]. The phenomenon of large bundles of neurons coordinating their electrical activity, rhythmically and repetitively depolarizing, is termed slow oscillatory neuronal activity. These pulsations range from 20 to 30 s and reflect important physiological restoration of the brain and blood oxygenation, precisely matching the time, rhythm and electrical activity of the N3 stage, confirming that this waste clearance system is primarily active during slow-wave sleep [7,12].

#### 3.2.3. Sleep and Alzheimer’s Disease

The most common dementia and a chronic age-related neurodegenerative disease, Alzheimer’s is associated with a deterioration of memory, language and the ability to self-care, among others [1]. The complex cascade of neurotransmitters and hormones involved in sleep regulation of the brainstem and hypothalamus is the same as that responsible for Alzheimer’s disease [14]. Sleep abnormalities such as insomnia and sleep apnoea are highly prevalent in patients with neurodegenerative disease, often predating the onset of cognitive or neurological impairment [8,14]. Although this finding suggests that sleep contributes to the onset of Alzheimer’s disease, the direction of causality is not clear. Sleep disorders may be connected to Alzheimer’s disease itself or associated factors such as pain, depression or drug therapy [14]. Major sleep disturbances include insomnia, sleep apnoea syndrome (SAS) and circadian rhythm sleep disorder. Sleep disturbances occur early in disease progression, with minor cases already demonstrating impaired sleep. Alzheimer’s patients have a shorter total sleep time, increased awakening and worse sleep efficiency compared to controls; specifically, slow-wave sleep was found to be impaired [14]. In Alzheimer’s disease, sleep-related issues appear to be associated with the suprachiasmatic nucleus, which regulates the circadian rhythm and naturally deteriorates with age [14]. Sleep disturbances are present in 25–35% of Alzheimer’s patients, often resulting in impaired slow-wave sleep, a shorter total sleep time and sleep fragmentation [14]. Despite this, however, only one third of Alzheimer’s patients suffer from clinically diagnosed sleep disturbances, placing some question over the causal interrelationship of sleep and neurodegenerative disease. Although only associations have been made so far, there are genetic links emerging between sleep and Alzheimer’s disease. One of the major genetic risk factors for Alzheimer’s disease is apolipoprotein E (APOE), of which reduced function is associated with sleep apnoea, the progression of sleep disturbances, cognitive performance and sleep deterioration [14]. APOE therefore provides a genetic link between sleep and neurodegenerative disease, adding to the validity of this relationship. We therefore suggest that sleep and sleep impairments likely play a compelling role in neurodegenerative disease in particular in regard to waste removal from the central nervous system [8,14,15]. It remains to be demonstrated whether other mechanisms aside from sleep universally impair this process.

#### 3.2.4. Sleep and Toxic Waste Products in Animals

Using position emission tomography (PET) scans and the tracer F-florbetaben, amyloid-beta levels in 20 mice were assessed during normal sleep and sleep deprivation; a one night comparison showed that there was a significant increase in amyloid-beta levels in the hippocampus and the thalamus in 19 out of 20 mice, demonstrating in vivo evidence of the effects of sleep deprivation on recognized neurodegenerative processes [15]. This relationship could be bidirectional, since amyloid-beta has also been linked to decreasing sleep quality. This can be attributed to the increase in interstitial space volume during sleep [2]. There is a doubling of amyloid-beta clearance in the sleep state, and conversely sleep deprivation shows a reduction in the clearance of CSF metabolites [3]. This indicates the usefulness of sleep monitoring as a non-invasive prognostic marker for neurodegenerative disease. The accumulation of amyloid-beta peptides within the brain parenchyma can lead to neuroinflammation and cognitive decline [16].

#### 3.2.5. Slow-Wave Sleep and Age

Sleep varies greatly across our lifespan, starting as polyphasic sleep during early life, becoming monophasic during childhood, then slowly decreasing in duration until the age of 60, when the amount of sleep either remains constant or increases [12]. During this time span, many micro and macro changes occur, the largest of which is the amount of slow-wave sleep, peaking during puberty and then declining with age [12]. The origin of the decrease in slow-wave sleep is still unknown, but a suggested mechanism is the neuronal loss occurring with age, as ageing is often accompanied by a significant loss in cortical grey matter, most of which occurs in the prefrontal cortex, where slow oscillations originate, according to the global hypothesis [12]. Alzheimer’s disease is often regarded as accelerated ageing. The gradual deterioration of slow-wave sleep over time is a possible explanation and would therefore result in less glymphatic clearance and an increased risk of neurodegeneration.

#### 3.2.6. Governors of Sleep

Although the timing and structure of sleep are controlled by both circadian rhythm and homeostatic processes, slow-wave sleep, R sleep, cortisol and melatonin levels are not affected by circadian rhythm and are mainly driven by homeostatic forces [2,7]. For example, the amount of slow-wave sleep increases with the number of waking hours [12]. Alongside these slow-wave oscillations, the neuromodulator norepinephrine has been found to regulate glymphatic clearance [1].

#### 3.2.7. The Chief Modulator: Norepinephrine

The level of arousal also plays an important role in the movement of CSF and ISF [2]. As we sleep, the central levels of norepinephrine decline (due to lowering locus coeruleus-derived noradrenergic tone), leading to the expansion of the extracellular space, decreasing resistance and therefore increasing CSF influx and ISF efflux [2]. Natural sleep is therefore associated with improved tracer penetration along the periarterial spaces and increased interstitial solute clearance, such as amyloid-beta [2]. These findings were recreated in anesthetized mice, with the volume fraction of the interstitial space during wakefulness being 13–15% and 22–24% during both sleep and anaesthesia, again suggesting that sleep eases convective fluid flow [1]. Additionally, norepinephrine receptor antagonists induced glymphatic clearance, suggesting norepinephrine release during the daytime could be suppressing clearance, by decreasing the amount of interstitial space [1]. This blockade of adrenergic signaling expanded the ISF volume, accelerated glymphatic clearance and was associated with slow-wave electroencephalograph activity [13]. Norepinephrine also suppresses choroid plexus CSF production [1]. These expansions and increases in CSF production decrease resistance and increase perfusion, leading to a further boost in the removal of metabolic waste products from the brain [1,7]. These findings indicate glymphatic clearance increase, its specificity to sleep and the link to levels of CSF flow, which can be modulated in a clinical setting.

#### 3.2.8. Heartbeat or Breathing Rate

During wakefulness, CSF exhibits a small-amplitude rhythm synchronized to the respiratory signal, whereas during sleep, within the N3 stage, large oscillations occur every 20 s, driven by ventricular movement, resulting in significantly greater inflow of CSF [7]. The physical forces propelling CSF in glymphatic clearance are intracranial pulsations. Intracranial pulsations have an established relationship with oscillations of blood pressure, which coincide with heart rate. As well as heart rate, lower-frequency events of respiration, such as vasomotion, were also demonstrated to contribute to glymphatic pulsations [13]. As opposed to systemic arterial pulsations dissipating at an arteriolar and venous level, in the brain, the rigid skull promotes propagation of arterial pulsations, which are still measurable in the microvasculature and venous outflow. Even acute decreases in arterial pulsations impair glymphatic clearance. Paradoxically, the N3 sleep stage which has the highest levels of CSF influx and amyloid-beta removal also has the lowest rates of arterial pulsations [13], suggesting that other factors are at play. Only recently, lower-frequency intracranial pressure oscillations produced by respiration were shown to complement cardiac pulsations, which could alternatively drive clearance. Ultrafast magnetic resonance imaging demonstrated that forced inspiration was a main driver of CSF flow in both the lateral ventricles and the subarachnoid space [13]. A combination of both heartbeat and respiratory rate appears to drive these pulsations.

### 3.3. Impaired Glymphatic Clearance

#### 3.3.1. Alzheimer’s Disease and Glymphatic Clearance

All prevalent neurodegenerative diseases are characterized by the accumulation of aggregated proteins [1]. Accumulations of amyloid-beta plaques and neurofibrillary tangles of hyperphosphorylated tau are implicated in the cognitive decline in Alzheimer’s disease [3]. Perivascular drainage pathways function as a sink for interstitial amyloid-beta and perivascular spaces are also associated with amyloid deposition and Alzheimer’s pathology [1]. Additionally, an abnormal perivascular space has been linked to impaired glymphatic clearance [1]. Amyloid-beta plays a role in synaptic regulation and neuronal survival. Interstitial bulk flow and amyloid-beta accumulation both occur in the perivascular space, but the predominant site of amyloid-beta accumulation is the cerebral arteries. Alongside this, abnormal enlargement of the perivascular space is a frequently observed difference between Alzheimer’s patients and healthy controls [1]. Since glymphatic clearance is responsible for the movement of tau and amyloid-beta aggregates out of the brain, glymphatic clearance is of utmost importance to neurodegenerative disease, but remains understudied [1].

#### 3.3.2. Alzheimer’s, Endfeet and AQP4

Glymphatic clearance is impaired in a rodent animal model of Alzheimer’s disease, due to changes in the number of AQP4 water channels responsible for the movement of CSF and ISF, expressed on astrocytic endfeet [3]. AQP4 is usually on the endfeet of astrocytes rather than the soma, with abnormal AQP4 localization associated with perturbed glymphatic clearance [17]. Since AQP4 polarity is crucial for CSF inflow and the clearance of amyloid-beta, the loss of AQP4 polarization in the brain contributes to the impairment of glymphatic function [18]. In the brain, AQP4 mainly exists in two isoforms: a long isoform, AQP4-M1, and a short isoform, AQP4-M23, which both form hetero-tetramers in the plasma membrane of astrocytic endfeet [17]. Furthermore, mouse models using AQP4 deletion showed a decreased clearance of amyloid-beta, confirming their involvement in neurodegeneration [2,3].

#### 3.3.3. Glymphatic Clearance and Age

Glymphatic clearance seems to clear toxic aggregates efficiently until the end of the reproductive lifespan, then the system seems to fail [1]. In old mice, a decrease in AQP4 expression, mis-localization of AQP4 away from the astroglial endfeet and reduced pulsations of the arterial wall led to a 40% reduction in amyloid-beta clearance from the brain (depicted in Figure 4.) [3]. Glymphatic activity in old mice was observed to be reduced by 80–90% [1,4]. This could explain the increase in frequency of amyloid-beta in aged brains. Amyloid-beta accumulation is increased due to impaired glymphatic clearance, but high levels of amyloid-beta in the interstitial space also impair fluid movement, creating a positive feedback loop, further reducing amyloid-beta deposition. This means patients with Alzheimer’s disease will mostly have impaired glymphatic clearance, which gradually gets worse. Although the loss of AQP4 polarization favors AD pathology, the cause and effect are not yet clarified [3]. In aged brains, the AQP4 channels on astrocytic endfeet relocate to the astrocytes’ soma due to astrogliosis, slowing the rate of CSF–ISF exchange [16]. Impaired glymphatic clearance was also observed in aged transgenic mice with amyloid plaques, but also in younger mice without plaques [3]. Interestingly, injection of amyloid-beta into CSF reduced glymphatic clearance. Higher amyloid-beta levels resulted in lower clearance of tau tangles [3]. In addition, breathing rates during sleep increase with age due to decreasing lung efficiency. These shallower breaths will decrease intracranial pressure and weaken glymphatic clearance. The strength of penetrating arterial pulsations also decreases with age [8].

### 3.4. Lifestyle Choices

No suitable drug has yet been developed for Alzheimer’s disease. Research has begun to emerge on individual lifestyle choices and their modulation of glymphatic activity. Behavioral interventions can be both preventative and curative and are frequently preferred over medication. Since we have established the link between glymphatic activity and the absence of suitable treatment, we next investgate an overview of lifestyle choices and their effects on glymphatic clearance, that may in turn exert an effect on neurodegenerative processes.

#### 3.4.1. Omega-3 Consumption

Marine-based fish oils known as omega-3 polyunsaturated fatty acids (n3-PUFAs) have been found to modulate glymphatic activity. Epidemiological studies associate increased levels of n3-PUFAs with lower incidence of neurodegenerative disease, and n3-PUFA supplementation has been suggested to delay or prevent the onset of Alzheimer’s disease [19]. The central nervous system is highly enriched with n3-PUFAs that exhibit potent anti-inflammatory properties [19]. High endogenous levels of n3-PUFAs improve impairment of spatial learning and memory induced by amyloid-beta. Both endogenous and exogenous n3-PUFAs promote amyloid-beta clearance and reduce aggregate formation by inhibiting the activation of astrocytes, protecting against the loss of AQP4 polarization, thus reducing the chance of amyloid-related injury [19]. They also exhibit anti-amyloidogenic activity, modulate aggregation and decrease downstream toxicity [19]. Dietary intake of n3-PUFAs improved cognitive decline in mild Alzheimer’s disease [19]. Supplementation demonstrates higher CSF influx and clearance, with AQP4 remaining polarized at the astrocytic endfeet, increasing the speed of glymphatic clearance [19]. AQP4 knockout mice exhibited no difference in glymphatic activity even with dietary n3-PUFA supplementation, indicating that AQP4 water channels are essential in n3-PUFAs’ improvement of glymphatic function [19]. n3-PUFA supplementation has therefore been suggested to delay or prevent the onset of AD, by improving glymphatic transport and decreasing amyloid aggregation [19].

#### 3.4.2. Intermittent Fasting

Intermittent fasting consists of cycles of fasting and then eating; a specific variation of this is alternate-day fasting, consisting of a day of eating followed by a fasting day for a number of days consecutively.

In the brain, AQP4 mainly exists in two isoforms: a long isoform, AQP4-M1, and a short isoform, AQP4-M23, which both form hetero-tetramers in the plasma membrane of astrocytic endfeet [17]. Intermittent fasting ultimately downregulates the expression of AQP4-M1, decreasing the AQP4-M1/AQP4-M23 ratio, and therefore increases AQP4 polarization along the paravenous space, boosting glymphatic clearance [17]. Intermittent-day fasting, i.e., alternatingly fasting on one day and then eating ad libitum the next, lowered the amount of amyloid-beta deposition. This fasting causes the liver to switch to fatty acid oxidation which increases the amount of β-hydroxybutyrate in the blood after 12 h [17]. β-hydroxybutyrate crosses the blood–brain barrier and acts as an endogenous histone deacetylase three (HDAC) inhibitor, which has been shown to exert a protective effect in Alzheimer’s disease progression [17]. HDAC inhibitors prevent histone acetylation, which regulates the expression of microRNA-130a, which represses the expression of AQP4-M1, changing the ratio between isoforms [17]. This endogenous HDAC inhibitor also increases the polarity of AQP4 expression on astrocytic endfeet, further increasing glymphatic clearance. Interestingly, the Alzheimer brain has significantly higher amounts of histone deacetylase 3, consistent with its involvement in neurodegeneration [17]. Importantly, cells exposed to amyloid-beta show a decrease in microRNA-130a expression and an increase in HDAC expression, creating a positive feedback loop that will antagonize the positive effect of intermittent fasting on the neurodegenerative process [17].

#### 3.4.3. Sleeping Position

Gravity affects the movement of blood and CSF through the brain, and therefore sleep position will likely play a role in the clearance of waste products from the brain [8]. Both intracranial pressure and cerebral hemodynamics are influenced by body posture [6], and patients with dementia were found to spend a much larger percentage of time in the supine position compared to controls, establishing an association between time in supine position and dementia [8]. An important factor in this clearance pathway is the stretch of nerves and veins in each position [6]. Glymphatic transport is most efficient in the right lateral sleeping position, with more CSF clearance occurring compared to supine and prone [6]. The average person changes sleeping position 11 times per night, but there was no difference in the number of position changes between neurodegenerative and control groups, making the percentage of time spent in supine position the risk factor, not the number of position changes [8]. The suggested mechanisms behind the effects of posture on clearance would appear to result from gravity and a restriction of venous drainage of the carotid veins. Unfortunately, detecting which position you spend most time in is only possible in a sleep laboratory, since self-reported sleep positions are often false [6].

#### 3.4.4. Alcohol Consumption

Alcohol can either boost or hinder glymphatic clearance, depending on dosage and whether consumption is chronic or acute. Alcohol’s effect on glymphatic clearance is independent of the wake or sleep state [18]. Prolonged amounts of excessive alcohol consumption were shown to have adverse effects on the central nervous system, with acute and chronic exposure to high doses of alcohol (1 g/kg) dramatically reducing glymphatic transport in awake mice [18]. Chronic exposure to high levels of alcohol increases GFAP expression, inducing the depolarization of AQP4 channels, but conversely decreasing the levels of inflammatory cytokines [18]. AQP4 polarity is crucial for CSF inflow and the clearance of amyloid-beta, potentially leading to alcohol-induced changes in these water channels to hinder glymphatic clearance [18]. Heavy alcohol consumption for prolonged periods of time greatly increases the risk of developing Alzheimer’s disease [18]. Intermediate alcohol consumption was also found to decrease glymphatic clearance for both acute and chronic usage [18]. Both intermediate and heavy dosage induced non-permanent changes in glymphatic activity, as after 24 h of sobriety, glymphatic function was fully restored [18]. In contrast, both acute and chronic exposure to low doses of alcohol (0.5 g/kg) increased glymphatic clearance, due to decreased GFAP expression, reducing the risk of Alzheimer’s disease [18].

#### 3.4.5. Exercise

Bulk glymphatic flow is accelerated by physical training and notably improves both memory and cognition in neurodegenerative disease [16]. Voluntary running over a six-week duration restored protein homeostasis in the brain, reduced inflammation by decreasing the activation of microglia and astrocytes, improved cognition and reduced the deposition of amyloid-beta through an increase in glymphatic clearance, but showed no effect on the blood–brain barrier permeability [16]. In addition to this, six weeks of physical exercise accelerated glymphatic clearance and reduced amyloid-beta accumulation by increasing the movement of ISF [16]. AQP4 expression in the cortex was also found to be higher along the paravascular space in the exercise group [16]. This improved AQP4 polarization and led to a decrease in both amyloid plaques and neuroinflammation [16]. These findings are consistent with the benefits of exercise on brain health and cognition in the elderly and demonstrates the usefulness of exercise as a neuroprotective lifestyle choice for brain aging and neurodegeneration [16]. According to the WHO, beneficial amounts of exercise consist of at least 150 min of moderate or 75 min of vigorous exercise a week [16].

#### 3.4.6. Chronic Stress

Chronic psychological stress is a common risk factor for Alzheimer’s disease. Short-term stress is crucial for adaptation and survival, but long-term stress can be detrimental to both body and mind [20]. Chronic stress accelerates the accumulation and deposition of amyloid-beta [20]. Mice exposed to stress exhibited decreased glymphatic influx and efflux, loss of AQP4 polarization and a reduction in AQP4-bearing astrocytes [20]. Stress triggers the hypothalamic–pituitary–adrenal (HPA) axis to release glucocorticoids. Alzheimer’s disease is associated with a dysfunctional HPA axis, demonstrated by high levels of cortisol in the blood [20]. Glucocorticoids act by binding to glucocorticoid receptors (GR) or mineralocorticoid receptors (MR) and decrease astrocyte numbers downregulating the number of AQP4 channels [20]. Stress increases the levels of glucocorticoids and therefore GR activation, which also trigger the amyloid precursor protein to form amyloid-beta [20]. Mifepristone, a GR antagonist, significantly improves impaired glymphatic clearance impaired by stress reversing AQP4 expression [20]. Mifepristone might therefore be a useful treatment for Alzheimer’s disease mediated through the glymphatic system [20].

## 4. Discussion

This paper provides a synthesis of currently tested lifestyle choices which could aid in preventing or slowing the progression of Alzheimer’s disease through increased glymphatic activity. The incidence of Alzheimer’s disease is rising, but there is currently no effective disease-modifying treatment. Similar to other neurodegenerative diseases, Alzheimer’s disease is characterized by the accumulation of aggregated proteins [1]; the accumulation of amyloid-beta peptides and tau within the brain parenchyma causes neuroinflammation, amyloid-beta plaques and tau tangles [16]. This deposition occurs along perivascular spaces. Glymphatic clearance acts within these spaces, moving tau and amyloid-beta aggregates out of the brain and thus reducing neurodegenerative processes [1].

Glymphatic clearance also offers an explanation for why dementias are generally age-related. In mice, clearance of misfolded proteins and other cellular debris is generally efficient but reduces in capacity over time and begins to fail at the end of the reproductive lifespan [1]. This was demonstrated by glymphatic clearance in old mice being reduced by 80–90%, and may at least partly explain the increased concentration of amyloid-beta in aged brains [4]. One suggested mechanism behind this is the loss of polarization of AQP4 water channels. AQP4 channels are usually polarized along astrocytic endfeet, but can lose polarization, becoming more evenly distributed around the soma and thus slowing the rate of CSF–ISF exchange [16,17]. Since AQP4 polarity is crucial for CSF inflow and the clearance of amyloid-beta, the loss of AQP4 polarization in the brain contributes to the impairment of glymphatic function [18]. AQP4 deletion results in decreased clearance of amyloid-beta, supporting its involvement in neurodegenerative processes [2,3].

The vast majority of glymphatic clearance occurs during sleep. There is a 90% reduction in glymphatic clearance during wakefulness and twice the amount of protein clearance from the brain intima during sleep [1]. During slow-wave sleep, delta oscillations are nested in high-voltage slow oscillatory neuronal activity, causing large bundles of neurons to harmonize, rhythmically and repetitively depolarizing over 20–30 s [6,7]. This increases CSF inflow within the interstitial cavities and boosts glymphatic activity, increasing interstitial solute clearance [2,6,7,8]. Sleep is a primary driver of bulk flow and is crucial in its modulation. These slow oscillations have been linked to sleep pressure, occurring in abundance early in the night and then decreasing over time [7]. Slow-wave sleep is linked to time spent awake, with an increase in waking hours increasing the amount of slow-wave sleep [12]. As well as within each night, sleep changes greatly across our lifespan. The percentage of slow-wave sleep is highest during puberty, and then declines with age, exacerbated in Alzheimer’s disease [12]. Age-related neuronal and cortical grey matter loss is thought to be responsible for the decrease in slow-wave sleep, particularly in the prefrontal cortex where slow oscillations are believed to originate [12].

The neuromodulator norepinephrine regulates sleep, but also glymphatic clearance. During sleep, the decrease in norepinephrine levels causes the expansion of the extracellular space, decreasing resistance and therefore increasing the rate of glymphatic clearance [2]. Norepinephrine also suppresses choroid plexus CSF production [1]. These expansions, together with the increase in CSF production, decrease resistance and boost perfusion, leading to a further increase in the removal of metabolic waste products from the brain [1,7]. Norepinephrine controls the overall quantity of solute clearance, but intracranial pulsations are the physical force that propel CSF along the parenchyma. Intracranial pulsations have an established relationship with oscillations of blood pressure, which coincide with heart rate. These disperse throughout the brain, aiding metabolism, and at the same time eliminate toxic waste products. Alongside heart rate, lower-frequency events of respiration, and vasomotion contribute to glymphatic pulsations [13].

Slow-wave sleep is linked to glymphatic clearance, but also dementia. A third of Alzheimer’s patients suffer from clinically diagnosed sleep disturbances, and the vast majority Alzheimer’s patients have a shorter total sleep time and impaired slow-wave sleep, with both these deteriorations of sleep often predating its onset [6,8,14]. The complex cascade of neurotransmitters and hormones involved in sleep regulation is affected in Alzheimer’s disease [14]. Additionally, in healthy mice, a single night of sleep deprivation was sufficient to increase amyloid-beta deposition [14,15]. Sleep impairment therefore appears as an influential risk factor for neurodegenerative disease [15] that should ideally be recognized by general practitioners and medical specialists alike.

In this manuscript, we have described the results of lifestyle choices on glymphatic clearance; we here provide a summary of the findings. Sleep position, alcohol intake, exercise, omega-3 consumption, intermittent fasting and chronic stress all modulate glymphatic clearance, thereby potentially altering the risk for Alzheimer’s disease. 1. Gravity affects the movement of blood and CSF throughout the brain, and sleep position will therefore play a role in the clearance of waste products from the brain [8]. Neurodegenerative patients spend a much larger percentage of time in the supine position, which suggests a connection between time in supine position and dementia [8]. Glymphatic transport is most efficient in the right lateral sleeping position, with more CSF clearance occurring compared to supine and prone [6]. 2. High levels of both endogenous and exogenous marine-based fish oils known as omega-3 polyunsaturated fatty acids (n3-PUFAs) are associated with lower incidence of neurodegenerative disease, and n3-PUFA supplementation has been suggested to delay or prevent the onset of Alzheimer’s disease [19]. n3-PUFAs promote amyloid-beta clearance and reduce aggregate formation by inhibiting the activation of astrocytes, protecting against the loss of AQP4 polarization, and therefore reduce the chance of amyloid-related injury [19]. They exhibit anti-amyloidogenic activity, decrease amyloid-beta production, modulate aggregation and decrease downstream toxicity [19]. 3. Alcohol consumption can either boost or hinder glymphatic clearance, depending on dosage and chronic or acute consumption. Acute and chronic exposure to high doses of alcohol (1 g/kg) dramatically reduces glymphatic transport in awake mice [18]. This suggests that heavy alcohol consumption for prolonged periods of time greatly increases the risk of developing Alzheimer’s disease [18]. On the other hand, both acute and chronic exposure to low doses of alcohol (0.5 g/kg) increased glymphatic clearance [18]. Low doses of alcohol improved glymphatic function, due to decreased GFAP expression, and avoided the loss of AQP4 [18]. Physical training in mice showed a notable improvement in both memory and cognition impairments, associated with neurodegenerative disease [16]. 4. Physical exercise decreased astrocyte and microglia activation, leading to reduced inflammation, and increased the movement of ISF [16]. The increase in ISF movement accelerated glymphatic clearance and reduced amyloid-beta accumulation [16]. The increase in ISF movement is due to improved polarization of AQP4, resulting in a decline in amyloid plaques and neuroinflammation [16]. This confirms the benefits of exercise on brain health and cognition in the elderly and demonstrates the usefulness for exercise as a neuroprotective lifestyle choice for brain aging and neurodegeneration [16]. 5. Chronic stress is a common risk factor for Alzheimer’s disease. Short-term stress is crucial for adaptation and survival, but long-term stress can be detrimental to both body and mind [20]. Chronic stress accelerates the accumulation and deposition of amyloid-beta [20]. Mice exposed to stress exhibited decreased glymphatic influx and efflux, decreased expression and loss of the polarization of AQP4 and a reduction in AQP4-bearing astrocytes [20]. 6. Intermittent fasting ultimately downregulates the expression of AQP4-M1, decreasing the AQP4-M1/AQP4-M23 ratio, and therefore increases AQP4 polarization along the paravenous space, increasing glymphatic clearance [17]. Intermittent fasting therefore improves cognitive function and decreases amyloid-beta deposition, by increasing polarity mediated through, and by the upregulation of, the AQP4-M1/AQP4-M23 ratio [17]. Intermittent-day fasting decreases the amount of amyloid-beta deposition.

These lifestyle choices in various ways modulate the levels of glymphatic clearance, lowering the risk of, or possibly even preventing, Alzheimer’s disease. Each lifestyle choice has a different mechanistic route, but all seem to function by changing the number or polarity of AQP4 channels. These can be split into two categories, with differing degrees of recommendations. Easily modifiable lifestyle choices include alcohol intake, omega-3 consumption, sleep position and exercise. Alcohol should only be consumed in low doses (0.5 g/kg) if at all, and avoided in moderate or high quantities. Omega-3 supplementation is recommended. Self-reported sleep position is often unreliable, however with the use of sleep positional therapies, an individual can be trained to alter their sleep position. Each individual should exercise moderately for 150 min a week or vigorously for 75 min. The less clear-cut lifestyle choices include intermittent fasting and chronic stress reduction. Intermittent fasting has only been investigated thus far in animal models, and can have harmful effects on humans, thus it requires further investigation. Chronic stress is treatable, but this is not as simple as taking supplementation and may require other therapeutic means. Although medication for chronic stress exists, this is a sensitive case-specific option that requires detailed and precise clinical assessment and is beyond the scope of this paper. Easily modifiable or not, lifestyle choices undoubtedly impact glymphatic clearance and should be harnessed to avoid brain ageing and neurodegeneration.

A limitation within this paper is the lack of direct information on Alzheimer’s disease sufferers. Although this literature review clearly highlights the effectiveness of lifestyle choices as a prevention of neurodegeneration, most of these findings come from murine studies, as there is only little in vivo human evidence for lifestyle choices altering neurodegeneration. Firstly, these findings need to be replicated in humans. Secondly, since this is emerging research and little literature exists, especially concerning lifestyle choices, each recommendation should be examined further before application. Therefore our suggestion that these lifestyle choices are causally linked is provided with the caveat that it is based on a small number of studies that need to replicated. Thirdly, these findings only demonstrate impaired glymphatic clearance, but the precise causal relationships still remain to be elucidated. Fourth and finally, these lifestyle choices need to be assessed in relation to each other, since we simply do not know whether these effects will summate, synergize or cancel each other out.

There seems a compelling need to capitalize on the glymphatic system to harness the potential for reducing dementia rates. Although AQP4 channels have been identified as a potential drug target, a suitable drug is yet to be developed; lifestyle choices therefore remain the best available option for regulating AQP4 numbers and polarization. Future studies should amongst others (see Table 1) empirically confirm the causality between lifestyle choices and improved glymphatic clearance, to quickly develop effective lifestyle interventions. Since existing glymphatic research consists mostly of animal studies, these findings need to be replicated in humans. Other avenues of research may include the role of exosomes (small extracellular vesicles) in transporting protein aggregates [21]; it is as yet unknown if disregulation of the glymphatic system may contribute to neurodegenerative processes by reducing exosome removal.

## 5. Conclusions

The glymphatic system is a brain-wide bulk flow waste removal network. Impaired glymphatic clearance contributes to the risk of developing Alzheimer’s disease, due to reduced removal of aggregated proteins such as amyloid-beta and tau. There is no effective treatment for Alzheimer’s disease, and with an increasing incidence and burden on society, the need for an intervention is well recognized. Lifestyle choices may be our current best option for reducing or even stopping the neurodegenerative process, as they constitute non-invasive, readily available, relatively inexpensive and accessible interventions.

## Figures and Tables

**Figure 1 brainsci-10-00868-f001:**
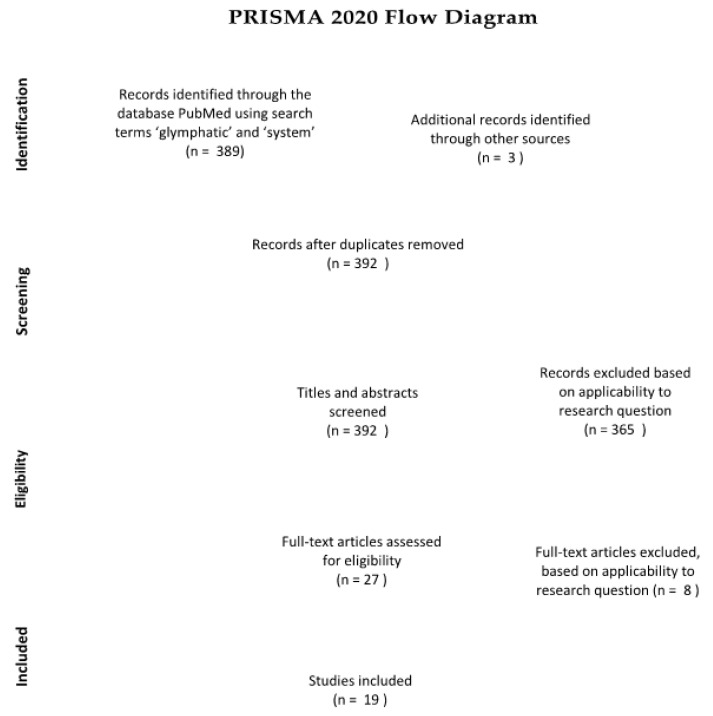
(Figure 1. was created according to the guidelines [9] p. 3). Of the 389 papers obtained, ultimately only 19 were included in the final literature review.

**Figure 2 brainsci-10-00868-f002:**
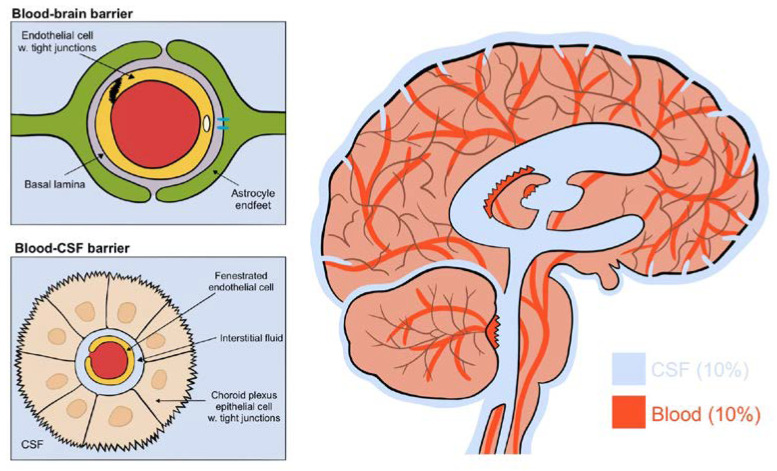
(Figure illustrated by Jessamyn Camille Reddy, adapted from; [1] p. 22, permission obtained) The four fluid compartments of the brain, separated by the blood–brain barrier or the blood–CSF barrier. The blood–brain barrier is situated wherever the vasculature reaches; the blood–CSF barrier is situated only in the choroid plexus and allows the passage of macromolecules.

**Figure 3 brainsci-10-00868-f003:**
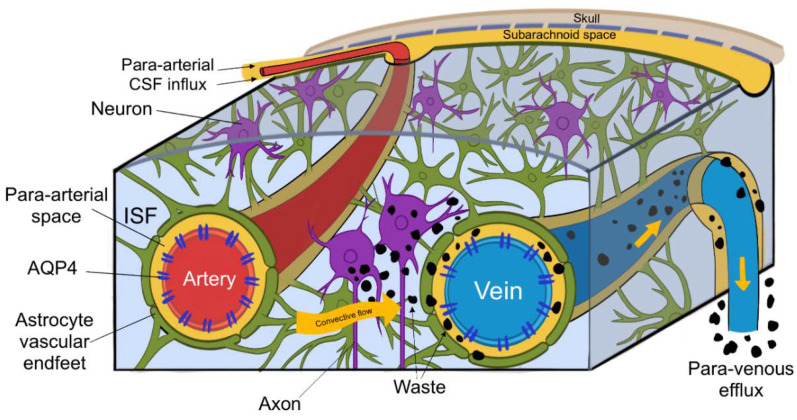
(Figure illustrated by Jessamyn Camille Reddy, adapted from; [2] p. 16) This figure depicts the circulation of cerebrospinal fluid (CSF) and its interchange with interstitial fluid (ISF), CSF entering the perivascular space of penetrating arteries, then through convective flow clearing waste products into the perivenous spaces, ultimately leaving the brain through paravenous efflux routes.

**Figure 4 brainsci-10-00868-f004:**
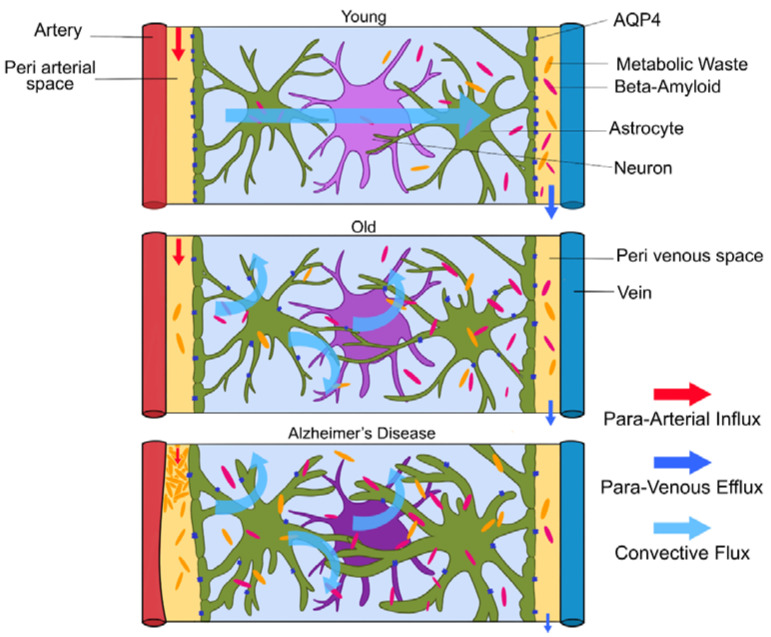
(Figure illustrated by Jessamyn Camille Reddy, adapted from; [1] p. 26, permission obtained) Model of glymphatic function in Young, Old and Alzheimer’s disease. In young people, CSF travels along periarterial routes, entering the brain parenchyma, and washes solutes and waste products into the veins. In older people, the loss of AQP4 water channels will result in reduced glymphatic clearance. In those with Alzheimer’s disease, the accumulation of amyloid-beta impairs fluid movement within the interstitial space, decreasing glymphatic clearance.

**Table 1 brainsci-10-00868-t001:** This table summarizes the future challenges arising from this paper.

Future Studies Surrounding the Glymphatic System and Lifestyle Choices should Aim to:
Confirm the link between the glymphatic system and lifestyle choices. Replicate all glymphatic-related murine research in humans. Test easily modifiable lifestyle choices affecting glymphatic clearance in a lifestyle-based intervention in humans. Investigate whether the effects of multiple lifestyle choices on glymphatic activity cumulate or cancel each other out. Confirm that AQP4 channels are the causal pathway behind glymphatic activity. Conduct a randomized controlled trial to confirm whether lifestyle choices have an effect on glymphatic activity. Investigate the effect of Mifepristone on glymphatic activity

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
