# Peer review of "The Sleeping Brain: Harnessing the Power of the Glymphatic System through Lifestyle Choices"

_brainsci, 2020, doi:10.3390/brainsci10110868_

Round 1

Reviewer 1 Report

This review is very topical and the authors have provided a reasonably comprehensive overview of the current field.

However, the manuscript requires considerable editing, especially in the discussion section which contains much simple duplication of earlier information.

The extent of the edits are too numerous to itemise individually. I have provided the editors with scanned copy of suggested edits within supplied manuscript for reference.

Author Response

Cover letter: The sleeping brain; harnessing the power of the glymphatic system through lifestyle choices

Oliver Cameron Reddy1 and Ysbrand D. van der Werf2*

Response to reviewers’ comments

We sincerely thank all the reviewers for constructive criticisms and valuable comments, which were of great help in revising the manuscript; and we were heartened to read the overall positive appraisal.

We conformed to all the suggestions made by the reviewers, concerning textual or typographical changes. All the changes were revised using Track Changes in Word. Another revision concerning the whole manuscript is that due to revisions and the reordering of the manuscript, the order of the references was changed. Therefore we decided to redo all the references to be sure that they are all correct. This included both the bibliography and the in-text citations. Due to the sheer scale of these changes, they were not recorded using track changes, because the edits would be too numerous. 

Below we detailed the larger, reviewer-specific changes. For all these revisions have indicated the line numbers. We will below elaborate on each reviewers feedback and our incorporation in more detail.

Reviewer 1

We sincerely thank you for your constructive criticisms and valuable comments, which were of great help in revising the manuscript. The document you provided us with was very detailed and elaborate, which was extremely helpful in bringing this manuscript up to standards.

Your comments were completely accurate, and a weakness of the initial drafts was simple duplications in the discussion section. We have deleted large parts of both the discussion (line 570-777) and conclusion (line 849-846), since there was much duplication. We have then also labelled the figures with the missing information, and changed aesthetic aspects such as font size and bullet point location (line 153 and 356). We have also added a brief introduction to the intermittent fasting paragraph (line 411-413) and the lifestyle choices section (line 386-391), which make the review much more readable. To accommodate another comment we deleted the sentence (line 375-377). The other smaller changes such as grammar issues are as you mentioned too numerous to cite here but we accommodated all and they are visible in the Track Changes.

Kind regards,

Oliver Cameron Reddy and Ysbrand D. van der Werf

Reviewer 2 Report

The authors wrote a highly interesting review on the role of the glymphatic system in Alzheimer's disease and its modulation by life choices. Thank you. The current manuscript is making a review of the sparse litterature on the field and help us in our understanding of the current research. They rise several points to be adress in the future regarding the validation of murine data in humans.

I have no major comments.

My only two comments are as follow:

1) I would really appreciate to see a simple table at the end of the paper to summurize the future challenges in the field since it is really recent.

2) I would like to see a paragraph in the review regarding the potential usage of the glymphatic system by exosomes (small extracellular vesicles). Central exosomes can be found at the periphery and can transport tau and Abeta proteins. The glymphatic system and its disregulation during aging and neurodegenerative diseases could contribute to disease progression indirectly by reducing exosome removal from the brain ? I am not sure that these data  have been published but would be happy to see a word in the review to open this question as future direction ?

Author Response

Cover letter: The sleeping brain; harnessing the power of the glymphatic system through lifestyle choices

Oliver Cameron Reddy1 and Ysbrand D. van der Werf2*

Response to reviewers’ comments

We sincerely thank all the reviewers for constructive criticisms and valuable comments, which were of great help in revising the manuscript; and we were heartened to read the overall positive appraisal.

We conformed to all the suggestions made by the reviewers, concerning textual or typographical changes. All the changes were revised using Track Changes in Word. Another revision concerning the whole manuscript is that due to revisions and the reordering of the manuscript, the order of the references was changed. Therefore we decided to redo all the references to be sure that they are all correct. This included both the bibliography and the in-text citations. Due to the sheer scale of these changes, they were not recorded using track changes, because the edits would be too numerous. 

Below we detailed the larger, reviewer-specific changes. For all these revisions have indicated the line numbers. We will below elaborate on each reviewers feedback and our incorporation in more detail.

Reviewer 2

We sincerely thank you for your constructive criticisms and two very valuable comments, which were of great help in revising the manuscript. We believe especially the table will provide an easy overview of where we believe the scientific community currently stands.

Concerning your first comment about the simple table at the end of the paper summarizing what we hope future research achieves, we agreed that this would be valuable and have added a table between the discussion and conclusion (line 847-848). Within this table we have listed seven aims which we believed future studies should aim to achieve.

 We thank you for your insightful comment on the possible role of exosomes, that we had overlooked. We have now added a sentence plus reference in the Discussion (lines 834-878) to this effect and feel it highlights the possible avenues for research even more.

All in all, your comments have been valuable contributions and have hopefully raised this manuscripts standards.

Kind regards,

Oliver Cameron Reddy and Ysbrand D. van der Werf

Reviewer 3 Report

This manuscript addressed the glymphatic system, its dysfunction and the major consequences of impaired clearance in order to link neurodegeneration and glymphatic activity with lifestyle choices. The authors investigated lifestyle choices which could aid in preventing or slowing the progression of Alzheimer’s disease.

The glymphatic system is a controversial theory by itself and it has been a hot topic for debates since its existence. This theory is quite recent and there are not many publications in the literature around this hypothesis. This review paper made an attempt to review all the literature related to lifestyle and brain clearance for the first time. Given the very sparse literature around glymphatics and lifestyle, this paper made a valuable contribution to summarise all the efforts in this field.

This is a well written and structured manuscript and easy to read. The authors have described well the literature in a systematic way. I believe that other researchers in this field can benefit from this review paper.

Author Response

Dear Reviewer, 

Thank you very much for your time and accepting to review this review. I would also like to thank you for your kind comments. I too hope that this will contribute to the scientific community. Since there are no improvements listed specifically, only minor spelling mistakes, I shall only be incorporating these into the updated version of the manuscript. Again, I am very please with you kind comments. 

Regards,

Oliver Reddy 

Round 2

Reviewer 1 Report

The revision for this manuscript have produced a document suitable that solves my problems.